# Pathogenesis, Diagnostic Challenges, and Risk Factors of Pott's Disease

Ira Glassman, Kevin H. Nguyen, Jane Giess, Cheldon Alcantara, Michelle Booth and Vishwanath Venketaraman *

College of Osteopathic Medicine of the Pacific, Western University of Health Sciences, Pomona, CA 91766, USA
* Correspondence: vvenketaraman@westernu.edu; Tel.: +1-909-706-3736

**Abstract:** Tuberculosis (TB) prevalence is increasing in developed nations and continuing to cause significant mortality in low- and middle-income countries. As a result of the uptick in cases, there also exists an increased prevalence of extrapulmonary TB. TB is caused by *Mycobacterium tuberculosis* (*M. tb*). When *M. tb* disseminates to the vertebral column, it is called Pott's disease or spinal TB. The frequency, symptoms, and severity of the disease range by the location of the spine and the region of the affected vertebrae. While the current literature shows that timely diagnosis is crucial to reduce the morbidity and mortality from Pott's disease, there is a lack of specific clinical diagnostic criteria for Pott's disease, and the symptoms may be very non-specific. Studies have shown that novel molecular diagnostic methods are effective and timely choices. Research has implicated the risk factors for the susceptibility and severity of Pott's disease, such as HIV and immunosuppression, poverty, and malnutrition. Based on the current literature available, our group aims to summarize the pathogenesis, clinical features, diagnostic challenges, as well as the known risk factors for Pott's disease within this literature review.

**Keywords:** *M. tb*; Pott's disease; spinal TB; HIV; vitamin D





## 1. Introduction

*Mycobacterium tuberculosis* (*M. tb*) is the second leading cause of death from an infectious agent worldwide [1]. In 2018, approximately 10 million people became infected with TB, and 1.5 million died due to TB [2]. Primary infection by *M. tb* often seeds itself in the lungs of its host, causing Tuberculosis (TB), though in rare instances, the infection may spread to the bone and joints. Skeletal TB comprises 10% of extrapulmonary TB cases, of which 50% involve the spinal column [3,4]. Spinal TB is otherwise known as Pott's disease, named after Pervical Pott, the first patient to present with classic spinal TB in 1779 [3,5].

The variety of the clinical manifestations of Pott's disease proves to be a significant challenge in providing a timely and correct diagnosis in both developed and developing countries. Despite advances in imaging modalities, the time between the initial onset of symptoms to the time of diagnosis ranges between four and six months [6,7]. A delay in diagnosis is the factor that most contributes to a worse prognosis with an increased likelihood of surgical treatment and neurological deficits [6–10]. Magnetic resonance imaging (MRI) is capable of identifying the soft tissue changes in spinal TB but is often employed at the time of neurological symptom presentation and is mistaken for nontuberculous spinal infection [3,11]. Once symptoms progress to neurological deficits, a significant number of patients may never recover their neurological function. Kumar et al. revealed that their surgical center noted that 28% of total pediatric cases remained the same or worse after surgery, with the majority of patients presenting with neurological symptoms [12]. Treatment at the early stages of Pott's disease with the standard treatment regimen leads to healing in approximately 95% of patients [13]. Therefore, early diagnosis is crucial to prevent the development of sequelae. However, physicians often exhibit a low index of suspicion of

Pott's disease, especially those from low-incidence countries [6]. Patients may present with non-specific symptoms and misleading data, such as a negative purified protein derivative (PPD) skin test despite active infection [3,14]. To avoid delays in diagnosis, there must be increased awareness of the diverse clinical presentations as well as improvements in the diagnostic criteria of Pott's disease.

In addition to identifying key diagnostic modalities for the detection of Pott's disease, it is crucial to understand the risk factors in patient populations to identify at-risk individuals and to expand upon the prophylactic measures to reduce the incidence of the disease. Immunodeficiency, most notably HIV, as well as vitamin D deficiency, are strongly implicated as risk factors associated with Pott's disease due to their profound influence on the body's ability to handle the infective processes of *M. tb* [15,16].

Pott's disease has surged in developed nations, especially targeting immunocompromised individuals, due to global migration [17]. It is paramount to identify at-risk individuals and diagnose them promptly to reduce the morbidity and mortality associated with this disease. It is important to shed light on this prominent health issue to increase attention and focus future research efforts in order to reduce the significant associated morbidity and mortality. The aim of this review is to review the current literature on the pathogenesis, diagnostic challenges, and risk factors of Pott's disease to aid clinical suspicion and elaborate on the need for further research.

## 2. Materials and Methods

To find studies for this article on the pathogenesis, diagnostic challenges, and risk factors of *M. tb*, a series of steps were performed. A literature search was conducted during September to December 2022 within the PubMed and NCBI databases. This included collecting data on keywords and inclusion and exclusion criteria. The search results included terms such as: "Tuberculosis", "*Mycobacterium tuberculosis*", "Pott's disease", "spinal tuberculosis", "diagnostic", "risk factors", and "pathogenesis". All relevant articles and their associated citations were reviewed. Attention was paid to each section to include articles published within the last 25 years, relevant *M. tb* and *M. tb* spinal infections. The exclusion criteria included non-relevance to *M. tb* and *M. tb* spinal infections, other extrapulmonary tuberculosis manifestations, and other causative nontuberculous mycobacterial spinal infections.

## 3. Pathogenesis and Clinical Presentation of Pott's Disease

With the recent uptick in the incidence of TB in both developed and developing countries, having an understanding of the rare presentations of TB is increasing in importance [3]. *M. tb* spreads through aerosol droplets to cause a primary infection in the lungs of the host due to its dependence on oxygen [18]. As seen in Figure 1, *M. tb* infection has multiple avenues in which it spreads throughout the host [19]. Without proper treatment or detection, the infection may become latent in the lungs or spread via hematogenous dissemination to the extrapulmonary sites of the body [18,20].

Spinal TB encompasses 2% of all TB cases, 15% of extrapulmonary cases, and 50% of skeletal TB cases [5]. Hematogenous dissemination is the primary mechanism by which spinal TB arises [20]. Spinal TB arises from a primary focus from the classic pulmonary focus or another extraosseous focus, such as the lymph nodes, GI tract, or another visceral location in which mycobacteria have infiltrated [5,18,20]. Approximately 67% of spinal TB cases are linked to primary pulmonary TB [4]. These foci could have active, subclinical, or quiescent disease [5,21]. For example, Jung et al. demonstrated a case of spinal TB originating from latent TB infection in a pediatric patient being treated for Crohn's disease with infliximab. The patient developed miliary TB one month after infliximab initiation and was subsequently cured of miliary TB; however, the patient developed spinal TB three months later [22].

**Figure 1.** Flow chart depicting the pathogenesis of Pott's disease. From primary infection, spinal tuberculosis may develop from hematogenous dissemination from the primary infection site and spread to an extrapulmonary site, one of which is the involvement of the skeletal system and, thus, the spine. Following this infection of the spine, spinal TB could present in four major presentations: paradiscal (involves the subchondral space and disc degeneration), central (stems from the center of the vertebral body), anterior/non-osseous (abscess formation between the spinal column and the anterior longitudinal ligament, often spanning multiple segments, sparing bone/disc involvement), and posterior (the infection seeds in the neural arch and posterior aspects of the vertebra). This leads to a compromise and the destruction of the spinal column and causes the classic and atypical presentations of spinal TB/Pott's disease [3,18–20].

After dissemination, the bacteria travel through the vascular system, using the anterior and posterior spinal arteries of each vertebra, to ultimately reach the cancellous bone region of the vertebra, with the most common site being the thoracolumbar junction [4,23]. Using the valveless Batson's paravertebral venous plexus, intrabdominal and intrathoracic pressures spread the infection to the inferior anterior portion of a vertebral body [5,18]. From here, the infection may utilize the anterior longitudinal ligament to infect adjacent vertebrae, as seen in Figure 2 [21]. Paradiscal and central are among the more common presentations of spinal TB, along with anterior/non-osseous and posterior lesions [4,5,20].

For paradiscal involvement, the most common type of lesion, *M. tb* lodges into the subchondral marrow of the vertebra leading to disc destruction [5,18,20]. This disc destruction leads to an anterior wedge of the involved vertebrae creating the characteristic kyphosis of Pott's disease. This type of spinal TB also often presents with intraosseous and extraosseous abscess formation, which places the patient at an increased risk for spinal cord damage and possible paraplegia [5]. These lesions are more likely to occur in younger patients due to the increased vascularization in pediatric paravertebral discs compared to adults [4].

For central involvement, *M. tb* infects the center of the vertebral body, leading to the destruction of the vertebral body itself, often sparing the disc [5,20]. This type of lesion leads to the complete compression of the vertebral body, known as vertebra plana [4]. Garg et al. demonstrated in a 5-year retrospective study of 1,652 patients in India that 82% of spinal TB patients had the paradiscal type of lesions, and 15.2% had the central type of lesions [24].

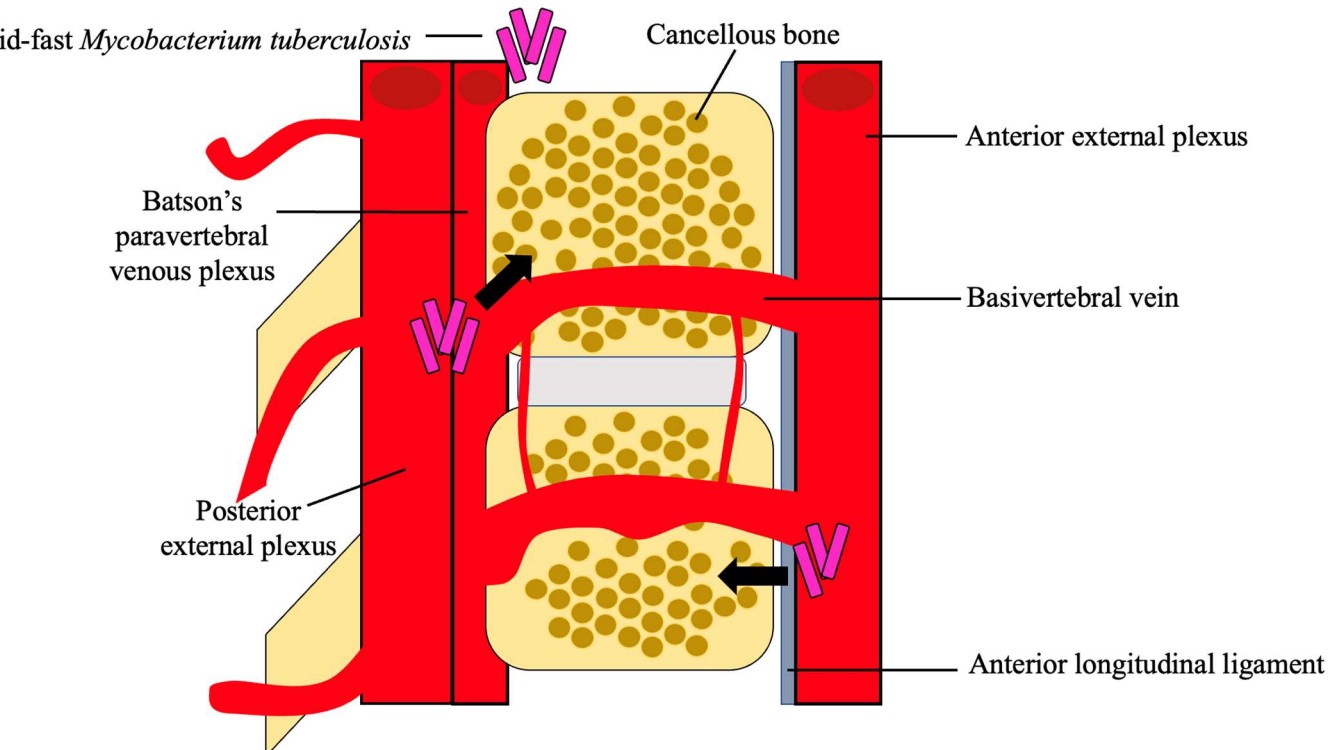

**Figure 2.** Hematogenous spread of *M. tb*. *M. tb* reaches the cancellous bone through the anterior and posterior external venous plexuses. A combination of the valveless Batson's paravertebral venous plexus and cavity pressures spreads *M. tb* to the inferior anterior portion of the vertebra and ultimately to the anterior longitudinal ligament where it may disseminate to adjacent vertebra [4,5,18,21].

Anterior or non-osseous involvement initially spares the bone and disc of the spinal column and creates an abscess using the anterior longitudinal ligament to spread over multiple contiguous vertebrae [5,18,20]. These abscesses are granulomatous in nature, and as they grow, the periosteum lifts, leading to bone devascularization and, eventually, necrosis and deformity [18,20]. Posterior involvement follows similar pathogenesis to anterior involvement yet utilizes the posterior longitudinal ligament and often involves the neural arch [18,20]. Paradiscal, central, and non-osseous-type lesions represent approximately 98% of all spinal TB cases, showing that posterior-type lesions are much less common [25].

Should the infection lead to bone collapse or spinal canal compromise, the infection would then be classified as TB spondylitis [26]. At this stage, the clinical findings may be hard to differentiate from a pyogenic or fungal osteomyelitis of the spine or metastatic bone tumors [26]. Given the potential level of destruction that spinal TB could lead to, it is important to understand the classical clinical presentation of the infection.

Spinal TB more frequently presents in children and young adults due to the increased vascularization of their spine, often with paradiscal lesions and in immunosuppressed patients [3,4]. Classically, spinal TB presents with pain and tenderness over the spine, neurological deficits, cold abscesses, fevers, and, if found in a later stage, a kyphotic spinal deformity and instability, as depicted in Figure 3 [3,5,20]. This presentation, however, depends on the duration of the disease, the severity of spinal destruction, and the site of the infection [3]. Spinal TB more frequently infects the lower thoracic and lumbar region but could present in the cervical spine as well [4,20]. Additionally, it typically presents without the symptoms of pulmonary TB, although both could be simultaneously present [20].

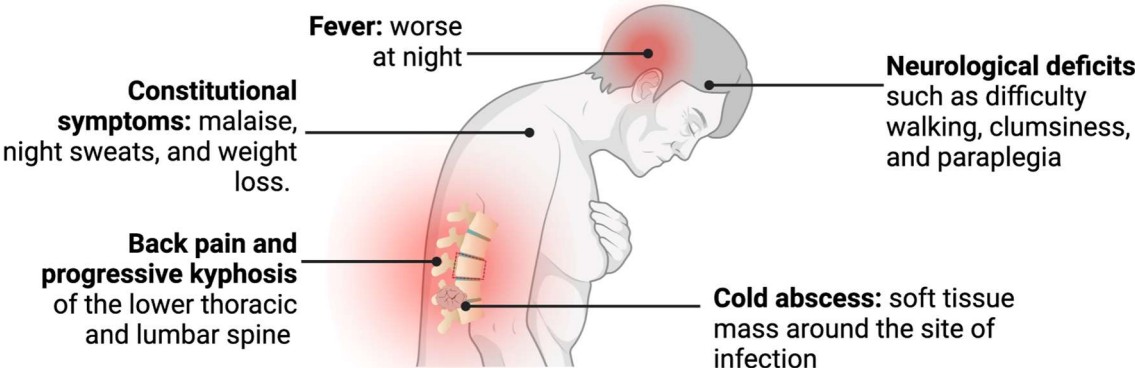

**Figure 3.** Pott's disease commonly presents in children or younger adults with back pain, neurological deficits, cold abscesses, and possible kyphotic deformities. Less commonly, patients present with constitutional symptoms and fevers, which generally precede the former symptoms [3–5,19,20,25,26].

The earliest and most common of these symptoms is back pain, which is associated with localized swelling and tenderness with progressive paraspinal muscle spasms, leading to restriction and pain in all planes of spinal motion [5]. In the later stages of the infection, back pain presents with an associated kyphosis of the affected spinal area [4]. This associated kyphosis could continue despite the resolution of the infection. In adults, kyphosis is limited to active disease, but in children, kyphosis could increase or worsen in periods of growth, further perpetuating other associated symptoms of Pott's disease, such as neurological deficits and possible paraplegia [3,20].

The second most common associated clinical symptom is neurological deficits, determined by the level at which the spine is affected [4]. Thus, an infection site in the cervical region would present with neurological deficits indicative of both upper extremity and lower extremity dysfunction, whereas a lumbar site of infection would only present with lower extremity and sacral deficits [4]. These deficits are a consequence of direct neural compression, the invasion of the neural parenchyma, tuberculous meningitis, pathological dislocation and the subluxation of the vertebrae, or vascular compromise to the spinal cord segment [18]. Since the infection starts at the anterior portion of the vertebrae, neurological deficits would work their way from anterior to posterior, starting with involvement of the anterior spinal tract, leading to exaggerated tendon reflexes and upper motor neuron-type deficits [3]. This could progress to a weakness of the limbs and difficulty with ambulation due to muscle spasms [25]. Left untreated, this further progresses to bladder and bowel dysfunction, sensory loss, and possible paraplegia [3].

Cold abscesses are also commonly associated with spinal TB and are typically located near the initial lesion [4]. A cold abscess denotes an abscess in which there are no inflammatory indications, such as warmth or erythema [3]. These abscesses may grow to be very large and add to the symptoms the patient presents with. For example, if the initial lesion is in the cervical region, the cold abscess could be formed in the retropharyngeal pouch and produce pressure effects that cause dysphagia, respiratory distress, or hoarseness of the voice [4].

Constitutional symptoms, such as malaise, fever, night sweats, and weight loss, are less common but still occur in a majority of patients [20]. These symptoms often present prior to the onset of spinal and neurological symptoms [18].

## 4. Diagnostic Challenges of Pott's Disease

Pott's disease has an insidious onset, with back pain as the most commonly presenting symptom [7,9,27,28]. A study by Pigrau-Serrallach et al. found that back pain was present in 83-100% of patients pooled across ten clinical studies, and only one-third of these patients had a fever or constitutional symptoms [9]. Kamara et al. found similar results, with 84% of patients with Pott's disease (n = 228) having localized back pain and 40% having fever [7]. The combination of non-specific symptoms with the slow onset and progression of the disease often draws physicians to other diagnoses.

In the early presentation of patients with Pott's disease, imaging, such as plain radiographs, CTs, or MRIs, is ordered. For a radiolucent lesion to be apparent on a plain radiograph, there should be 30% of bone mineral loss [5]. Therefore, it is difficult to diagnose early Pott's disease using a plain radiograph. It was found that they could accurately diagnose Pott's disease in half the cases, indicating low specificity [27]. CT is typically used as a means of guidance for biopsy [5]. It is widely documented that MRI is superior to plain radiographs in the diagnosis of Pott's disease [4–6,9,13,27–31]. Spinal MRI determines the extent and nature of the bony destructions as well as the soft tissue's involvement [4]. The typical MRI pattern of Pott's disease shows vertebral bone destruction with relative preservation of the disc [9]. However, Wang et al. asserted that MRI does not help differentiate Potts disease from pyogenic vertebral osteomyelitis [11]. Overall, MRI is the ideal imaging modality for the early detection of this condition.

The initial suspicion of Pott's disease is often based on the clinical presentation and imaging data, but the definitive diagnosis involves the isolation and identification of the pathogen [9]. The typical laboratory procedure for clinical specimens involves an acid-fast bacilli (AFB) stain, the isolation of the organism by culture, and the identification and drug susceptibility of the recovered organism. A study by Wang et al. found that Pott's disease was excluded as a diagnosis in patients who were eventually found to be positive. This was mainly based on negative tests of *M. tb* without consideration of the extremely high false negative results [4,6,11]. This was demonstrated in a case report by Canine et al. when an AFB stain of abscess fluid was negative, but 10 days later, upon readmission, a chart review showed a culture positive for *M. tb* and a positive QuantiFERON-TB Gold serum test from the same initial epidural abscess [29]. Pandita et al. found that in contrast to pulmonary TB, the bacilli population in extrapulmonary tissue biopsy samples is much lower due to the paucibacillary state of the bacteria in less oxygenated osseous tissue [32]. None of the studies published to date showed that the microscopic examination positivity of the vertebral or paravertebral samples exceeded 36% [6]. In concordance with the literature, it is imperative that negative microbiologic results should not exclude the diagnosis of Pott's disease [4,9,11,27,29,30].

The erythrocyte sedimentation rate (ESR) and C-reactive protein (CRP) have historically been used as biomarkers for spinal tuberculosis. However, these non-specific inflammatory markers could remain normal in some patients or become elevated in response to an auxiliary infectious process [33]. Recent research has attempted to relieve some of the diagnostic ambiguity surrounding spinal TB by isolating more precise individual biomarkers and biomarker combinations. In August 2022, Lou et al. showed that lipopolysaccharide-binding protein (LBP) expression was significantly elevated in the peripheral blood smears of one hundred spinal TB patients [34]. LBP levels correlated positively with the ESR and CRP levels, meaning LBP may be a promising potential spinal TB biomarker [34]. Furthermore, in a 26-patient case-control study, Mann et al. investigated 19 biomarkers cited in pulmonary TB literature and determined that fibrinogen, IFN-y, NCAM, and ferritin, in addition to CRP, had the highest discrimination quality in differentiating patients suffering from spinal TB from those with mechanical back pain [33]. While CRP was confirmed as the most accurate individual serologic marker, a five-biomarker signature comprised of CRP, NCAM, ferritin, CXCL8, and GDF-15 was able to correctly distinguish spinal TB patients regardless of HIV status [33]. Using a multi-biomarker signature in lieu of ESR and CRP levels, which fluctuate both between patients and over the

disease course, could significantly improve spinal TB diagnostic accuracy in heterogenous populations [33]. However, larger prospective cohort studies are required to assess the accuracy of the biomarker candidates particularly among HIV positive individuals before their introduction into the clinical setting.

Recent developments in molecular methods, specifically the Xpert MTB/RIF, significantly reduce the overall time to obtain results with high sensitivity and specificity. Multiple studies have supported the high sensitivity, specificity, and rapid diagnosis of extrapulmonary TB [5,9,28,32,35]. Vadwai et al. found that the overall sensitivity and specificity for smear-positive cases in patients with suspected extrapulmonary TB were 96% and 99.6%, respectively [36]. Tortoli et al. found an overall sensitivity and specificity of 81.3% and 99.8%, respectively. For Xpert, the microscopy sensitivity was 48% [37]. In another study by Boehme et al., the MTB/RIF test reduced the median time to treatment for smear-negative TB from 56 days to 5 days [38]. Integrating Xpert MTB/RIF with current practices would significantly increase bacteriological confirmation, reducing the time to diagnosis and treatment.

A major obstacle to timely diagnoses is the lack of specific clinical diagnostic criteria for Pott's disease. The diagnostic criteria should emphasize that the presence of inflammatory back pain, even in the absence of constitutional symptoms, TB risk factors, or initial negative TB cultures, should suggest Pott's disease as a differential diagnosis [4,11,29]. If there is any suspicion of Potts disease, relevant lab testing, including the Xpert MTB/RIF test, as well as MRI imaging, should be performed to ensure an early diagnosis.

## 5. Risk Factors for Potts Disease

Human immunodeficiency virus (HIV) is one of the primary risk factors for developing Pott's disease, often coexisting with spinal TB in the endemic areas of Sub-Saharan Africa, to significantly complicate diagnosis and management [39]. Sub-Saharan Africa is home to 70% of the total number of HIV-positive patients in the world, and of the 10.4 million new cases of TB in 2015, 11% were among HIV-positive patients [39]. HIV and *M. tb* potentiate one another, with co-infection increasing both the severity of TB symptoms and the likelihood of progressing from HIV to AIDS [15]. The depletion of CD4+ T-cells by HIV is thought to have a major role in increasing TB co-infection risk [40]. Recent data from Lu et al. showed that CD4+ T-cells are vital for host immunity to *M. tb* infection by enhancing CD8+ T-cell effector functions and preventing CD8+ T-cell exhaustion, which plays a crucial protective role [41]. The Th-1 subtype of CD4+ T-cells releases IFN-$\gamma$, a cytokine that activates inflammatory macrophages to promote the phagocytosis of TB-infected macrophages [40]. The depletion of Th-1 CD4+ T-cells and the lack of IFN-$\gamma$ increases the incidence of initial TB infection, latent disease reactivation, and the extrapulmonary manifestations seen in Pott's disease [42]. *M. tb*, in turn, increases the expression of CXCR4 receptors to favor HIV replication and increase the viral load [43]. Poverty, poor access to care, and bias against HIV-positive individuals drastically increase the risk of HIV and TB co-infection, particularly in the developing countries of Sub-Saharan Africa [39].

Vitamin D deficiency was also correlated with increased susceptibility to Pott's disease, specifically caseous necrosis-type spinal TB, as well as an increased likelihood of necrosis compared to individuals with normal vitamin D levels [16]. Panwar et al., Agarwal et al., and Tang et al. showed that serum 1,25-dihydroxyvitamin D levels are significantly reduced in patients with spinal TB [16,44,45]. Active vitamin D is thought to enhance innate immunity by promoting the fusion of macrophage and phagolysosome complexes, mediating reactive oxygen species (ROS) generation, and reducing peroxisome proliferator-activated receptor expression to inhibit lipid metabolism in TB-infected macrophages [16]. Upon initial infection with *M. tb*, alveolar macrophages (AMs) are the first responders in generating an immune response [46]. While much of the current literature on the initial M. tb infection stems from murine models, it is evident that *M. tb* could bypass more permissive AMs, such as those with reduced ROS generation due to vitamin D deficiency, to grow

intracellularly [47]. Active vitamin D also enhances adaptive immunity by promoting FoxP3+/IL-10+ Treg cell differentiation and boosting T-cell immune tolerance [16].

Reduced serum levels of active vitamin D may be attributed to malnutrition, particularly in Sub-Saharan Africa, where even 50.06% of apparently healthy children experience vitamin D deficiency [48], but also to vitamin D receptor (VDR) gene polymorphisms, liver and renal failure, and estrogen deficiency. Zhang et al. showed that the FokI polymorphism in VDR genes, specifically the ff genotype, increases spinal TB susceptibility in Han Chinese populations [49]. The FokI polymorphism creates a new start codon on the second exon of chromosome twelve, yielding a shorter VDR protein with a higher transcriptional activity that depletes serum 1,25-dihydroxyvitamin D levels [50]. Patients with liver and renal failure, on the other hand, have intact VDR receptors but issues with activating vitamin D. Ingested vitamin D is enzymatically converted to 25-dihydroxyvitamin D in the liver and then to its active form 1,25-dihydroxyvitamin D in the kidney [51]. Patients with liver and renal failure lose the ability to perform these conversions, resulting in diminished levels of active vitamin D and an increased susceptibility to Pott's disease. Li et al. also found a connection between the VDR Bsml polymorphism and an increased risk of osteoarthritis, which may increase the risk that joints are damaged and may be more susceptible to infection [52]. Likewise, estrogen increases the conversion of 25-dihydroxyvitamin D to 1,25-dihydroxyvitamin D, serving to maintain bone mineral density [53]. In estrogen deficiency, activated T-cells secrete RANKL, TNF, and IL-17A, which increase bone resorption, reducing the bone mineral density and increasing the vertebral interstitial space, a potential factor that could increase the susceptibility to spinal infections [54,55]. While researchers have yet to link estrogen deficiency and spinal TB explicitly, Nhamoyebonde et al. identified a male bias in pulmonary TB incidence and suggested the protective effect of female estrogen as a possible explanation [56]. As such, the connection between estrogen deficiency and spinal TB merits further study.

In addition to the risk of susceptibility, certain patient populations are at risk for a delayed diagnosis. This delay could be due to false-negative PPD skin reactions due to a reduction in the delayed-type hypersensitivity response [57]. These populations include individuals with malnutrition or elderly populations, as evidenced by Maron et al. [14]. There is also evidence that aging yields more permissive AMs, such as the increased susceptibility seen in vitamin D deficiency, as evidenced by Lafuse et al. in their discovery of more susceptible aged AM populations in murine models [58]. Poverty, malnutrition, and poor access to care yield highly susceptible populations with insufficient means to promptly diagnose cases.

## 6. Conclusions

Pott's disease arises from the extrapulmonary dissemination of *M. tb* to cancellous bone in the vertebrae through the anterior and posterior venous plexuses, ultimately spreading to adjacent vertebrae, which yield the characteristic back pain and neurological deficit symptoms. The understanding of clinical presentations helps to guide suspicion for diagnosis; however, confirmation of the pathogen may prove challenging. MRI is the superior imaging modality for early detection to visualize soft tissue involvement. Despite AFB stain being the gold standard in identifying *M. tb*, extrapulmonary *M. tb* samples have a lower yield of the specimen and, thus, a lowered sensitivity as well as a lengthy culture time. Xpert MTB/RIF represents a molecular diagnostic technique that significantly improves diagnostic timing and boasts high sensitivity and specificity, as well as provides clues to antimicrobial resistance to further guide individual patient treatment plans. In addition to molecular techniques and imaging, novel biomarker screens have proven to be effective in detecting *M. tb* infection, even in immunocompromised populations such as those with HIV. Research should continue to focus on molecular diagnostic methods for identifying difficult-to-diagnose bone and joint infections such as Pott's disease. Current research has shown that *M. tb* spreads most easily in individuals who are immunocompromised or vitamin D deficient. In addition to the increased susceptibility, these patients are at risk of

increased severity from infection due to impaired macrophage responses and diminished bone mineral densities. This ultimately results in disproportionately at-risk populations due to poverty, especially when coupled with delayed diagnoses due to inadequate access to care and novel detection mechanisms. More research is needed to investigate the role of estrogen deficiency as a risk factor for male dominance, as well as female patients who may be taking estrogen blocker therapy, i.e., for preventing breast cancer recurrence.

**Author Contributions:** Conceptualization, V.V. and I.G.; methodology, I.G.; resources, I.G.; writing—original draft preparation, I.G., J.G., M.B., C.A. and K.H.N.; writing—review and editing, I.G. and K.H.N.; visualization, J.G. and K.H.N.; supervision, V.V. and I.G.; project administration, V.V. and I.G. All authors have read and agreed to the published version of the manuscript.

**Funding:** This research received no external funding.

**Institutional Review Board Statement:** Not applicable.

**Informed Consent Statement:** Not applicable.

**Data Availability Statement:** Not applicable.

**Acknowledgments:** We appreciate the funding support from NIH (HL143545-01A1). Figures created with BioRender.com.

**Conflicts of Interest:** The authors declare no conflict of interest.

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
