# Peer review of "Pathogenesis, Diagnostic Challenges, and Risk Factors of Pott’s Disease"

_clinpract, doi:10.3390/clinpract13010014_

Round 1

Reviewer 1 Report

Comments

The authors highlight the pathogenesis part in the manuscript, but when it comes to the diagnostic challenges, they only discuss the limitations of existing diagnostics without discussing the scope of improvements and other diagnostic variants that are currently being explored or studied. For example, biosensors employing critical biomarkers of TB can be discussed in the review to provide a broader perspective for readers. A discussion of only the challenges of this disease will not provide readers with a comprehensive overview; instead, an explanation of future prospects and current developments/developments in pipeline would be more relevant.

·      Therefore, the author should discuss the currently available biomarkers for spinal TB. These are very well presented in articles like https://www.ncbi.nlm.nih.gov/pmc/articles/PMC8637108/ and https://pubmed.ncbi.nlm.nih.gov/36060237/

·      And secondly, the implications of these biomarkers for the development of clinical biosensors for extra-pulmonary tuberculosis must be included as they are gaining tremendous attention nowadays, for which I would recommend adding the review, https://pubmed.ncbi.nlm.nih.gov/35196464/ , wherein they are very nicely discussed

·      Reference section:

The references need to be in a proper format and easier to interpret; some are even intermixed. I recommend that the authors follow the journal's author guidelines for references.

Author Response

Dear Reviewer #1,

We appreciate the time and effort you have all taken to provide an extensive and thorough review of our manuscript. We are also very encouraged by how well it was accepted, and we agree that the suggested edits have greatly improved our manuscript. The objective of our paper was to provide an overview of Pott’s disease pathogenesis and presentations, diagnostic challenges, and risk factors for the disease. We hope that our revised manuscript based on all of the suggested edits make this objective much clearer.

Thank you all for you valuable time.

Reviewer #1:

The authors highlight the pathogenesis part in the manuscript, but when it comes to the diagnostic challenges, they only discuss the limitations of existing diagnostics without discussing the scope of improvements and other diagnostic variants that are currently being explored or studied. For example, biosensors employing critical biomarkers of TB can be discussed in the review to provide a broader perspective for readers. A discussion of only the challenges of this disease will not provide readers with a comprehensive overview; instead, an explanation of future prospects and current developments/developments in pipeline would be more relevant.

  • Therefore, the author should discuss the currently available biomarkers for spinal TB. These are very well presented in articles like https://www.ncbi.nlm.nih.gov/pmc/articles/PMC8637108/ and https://pubmed.ncbi.nlm.nih.gov/36060237/

  • And secondly, the implications of these biomarkers for the development of clinical biosensors for extra-pulmonary tuberculosis must be included as they are gaining tremendous attention nowadays, for which I would recommend adding the review, https://pubmed.ncbi.nlm.nih.gov/35196464/ , wherein they are very nicely discussed

Response: Thank you very much for this suggestion. This was a great idea, and we believe greatly enhanced the quality of our review. We have added to the diagnostic challenges section information on current developments in biomarkers and their improvements on existing laboratory markers. We also suggested further studies and tied into the immunosuppression relevance to flow into the next section.

  • Reference section:

The references need to be in a proper format and easier to interpret; some are even intermixed. I recommend that the authors follow the journal's author guidelines for references

Response: Thank you. We have corrected the citation format and the grouped citations were fixed.

Reviewer 2 Report

The manuscript “Pathogenesis, Diagnostic Challenges, and Risk Factors of Pott’s Disease “ is well written and comprehensive review of Pott’s disease. Though the authors mention treatment in abstract and few lines on management of Potts disease in the Introduction, there is further description of the same. The authors could mention briefly on treatment modalities/management with specific MTB treatment as well as any surgical intervention. Additional information of morbidity associated with delayed diagnosis could be emphasized. If they chose not to add treatment, then the lines on management of Pott’s disease in Introduction can be reduced and more focussed on pathogenesis, diagnosis and risk factors.

Minor Comments-

Lines 42-43 It would be worthwhile to add a note on paediatric cases and t heir prognosis

Lines 71-72 In Figure 1 or text could you add approximate % who gets active disease, latent infection a, dissemination, as well as distribution of each type of spinal TB (predominance of each as seen later in lines 98) and so on….

Also, you could avoid the word “etc” in figure

Figure 2 – The labels can have either sentence case or Normal case uniformly.

Line 89 ‘Hematogenous dissemination is the primary mechanism in which spinal TB’. Is there any role for reactivation of latent TB? Any risk groups for the same?

Line 184 – “Diagnostic Challenges of Pott’s Disease”. The authors could give a list of diagnostic test; Imaging techniques. Laboratory parameters as well as MTB detection laboratory methods with their sensitivity and relative time to diagnosis of each test.

Line 214 - Pandita et al. found that in contrast to pulmonary 214 TB, the bacilli population in extrapulmonary samples is much lower – Please provide citation. Please specify lower by what test?

Line 237 – “Risk Factors for Potts Disease”. Is the risk factor associated with severity of Potts disease. If so, add a note on the same.

Author Response

Dear Reviewer #2,

We appreciate the time and effort you have all taken to provide an extensive and thorough review of our manuscript. We are also very encouraged by how well it was accepted, and we agree that the suggested edits have greatly improved our manuscript. The objective of our paper was to provide an overview of Pott’s disease pathogenesis and presentations, diagnostic challenges, and risk factors for the disease. We hope that our revised manuscript based on all of the suggested edits make this objective much clearer.

Thank you all for you valuable time.

Comments from Reviewer#2:
The manuscript “Pathogenesis, Diagnostic Challenges, and Risk Factors of Pott’s Disease “ is well written and comprehensive review of Pott’s disease. Though the authors mention treatment in abstract and few lines on management of Potts disease in the Introduction, there is further description of the same. The authors could mention briefly on treatment modalities/management with specific MTB treatment as well as any surgical intervention. Additional information of morbidity associated with delayed diagnosis could be emphasized. If they chose not to add treatment, then the lines on management of Pott’s disease in Introduction can be reduced and more focussed on pathogenesis, diagnosis and risk factors.

Response: Thank you very much for this feedback. We agree that the verbiage regarding treatment began to hint at treatment more than we wished. We have erased some of the wording regarding treatment to retain the focus on presentation, diagnostic challenges, and risk factors. We have left in one line regarding treatment only to shed light on the importance of early detection for significance.

Minor Comments-

Lines 42-43 It would be worthwhile to add a note on paediatric cases and t heir prognosis

Response: We have added a paper from a surgical center which noted that their pediatric patient population presenting with neurological symptoms yielded poor outcomes.

Lines 71-72 In Figure 1 or text could you add approximate % who gets active disease, latent infection a, dissemination, as well as distribution of each type of spinal TB (predominance of each as seen later in lines 98) and so on….

Response: Thank you for this suggestion. We have added several lines which include statistics we were able to find on spinal TB and predominance of subtypes.

Also, you could avoid the word “etc” in figure

Response: We have removed “etc” in figure.

Figure 2 – The labels can have either sentence case or Normal case uniformly.

Response: We have switched to sentence case for this figure.

Line 89 ‘Hematogenous dissemination is the primary mechanism in which spinal TB’. Is there any role for reactivation of latent TB? Any risk groups for the same?

Response: We have added a statistic as well as a case example of latent TB yielding spinal TB infection.

Line 184 – “Diagnostic Challenges of Pott’s Disease”. The authors could give a list of diagnostic test; Imaging techniques. Laboratory parameters as well as MTB detection laboratory methods with their sensitivity and relative time to diagnosis of each test.

Response: Thank you for this suggestion. This is something we initially sought to achieve in the paper, however data regarding specific sensitivity/specificity for Pott’s disease is much more limited, as much of the data pertains to tuberculosis infection itself.

Line 214 - Pandita et al. found that in contrast to pulmonary 214 TB, the bacilli population in extrapulmonary samples is much lower – Please provide citation. Please specify lower by what test?

Response: Thank you for catching this. The citation has been added and the test is specified: it was cell count by biopsy.

Line 237 – “Risk Factors for Potts Disease”. Is the risk factor associated with severity of Potts disease. If so, add a note on the same

Response: Thank you for this suggestion. This inspired us to think more about risk factors beyond susceptibility and amended the section to include information regarding severity as well.

Reviewer 3 Report

We congratulate the authors of this manuscript for the job done

Although this is just a general review of the literature which offer little benefit with respect to advancement of knowledge when compared to a systematic review with structured and more systematic collection of data, it has however provided reminder and useful evidence that could aid physician and other healthcare workers managing patients with Potts disease

Abstract

This need to be improved to better capture the main theme and message of the review article

Introduction

One possible improvement for this introduction would be to provide more context and background information about Pott's disease, including its prevalence and impact on global health. It could also be helpful to provide a brief overview of the objectives and main points of the review, and to clearly define any important terms or concepts that will be discussed. Additionally, it might be useful to provide more specific information about the diagnostic challenges and risk factors associated with Pott's disease, and to explain how these factors contribute to delays in diagnosis and treatment. Finally, it could be helpful to provide some context for the relevance of this review, such as the need to improve awareness and understanding of Pott's disease in order to reduce morbidity and mortality.

Method

One possible improvement for the method section of this review article could be to provide more detail and clarity about the search strategy and inclusion/exclusion criteria. This could include specifying the specific databases that were searched (such as PubMed and NCBI), the time frame of the search (e.g. articles published in the past 10 years), and the specific keywords and MeSH terms that were used. It could also be helpful to provide more information about how the inclusion and exclusion criteria were applied, such as how studies were selected for inclusion and how duplicates were identified and removed. Additionally, it might be useful to provide some information about the characteristics of the studies that were included in the review, such as the study design, sample size, and patient population. This would help to provide more context and transparency about the studies that were included in the review and how they were selected. Also should also state here the period of the articles covered, quality assessment and how bias was addressed in article selection

Pathogenesis and Clinical Presentation of Pott’s Disease

One possible improvement for this section of the review article on the pathogenesis and clinical presentation of Pott's disease could be to provide more detail and clarity about the various presentations of spinal TB and the mechanisms by which they develop. This could include more information about the specific areas of the spine that are affected in each presentation and the specific characteristics and symptoms that are associated with each presentation. It could also be helpful to include more information about the diagnostic challenges and pitfalls that may be encountered when evaluating patients with Pott's disease, as well as any specific considerations that may need to be taken into account when formulating a treatment plan. Additionally, it might be useful to provide more information about the prognostic factors and outcomes associated with each presentation of spinal TB, including the likelihood of complications and disability and the overall effectiveness of different treatment approaches. Citing relevant studies and clinical guidelines could also be helpful in supporting the points made in this section of the review.

Risk factors for Pott disease

Mentioning other risk factors for Pott's disease, such as older age, malnutrition, and previous TB infection

Providing more information on the role of vitamin D deficiency in the development of Pott's disease, including mechanisms by which it may increase susceptibility

Discussing the prevalence of Vitamin D deficiency in populations at risk for Pott's disease, such as those living in Sub-Saharan Africa

Clarifying the relationship between estrogen deficiency and Pott's disease, including providing more information on the possible protective effect of estrogen on TB incidence

Providing more detail on the link between HIV and Pott's disease, including how HIV infection may increase the risk of developing Pott's disease and how Pott's disease may contribute to HIV progression

Expanding on the role of genetics in the development of Pott's disease, including the potential role of other gene polymorphisms in addition to the FokI polymorphism in the VDR gene

Mentioning the impact of poverty and poor access to care on the risk of developing Pott's disease and the challenges these factors pose in diagnosis and treatment.

Author Response

Dear Reviewer #3,

We appreciate the time and effort you have all taken to provide an extensive and thorough review of our manuscript. We are also very encouraged by how well it was accepted, and we agree that the suggested edits have greatly improved our manuscript. The objective of our paper was to provide an overview of Pott’s disease pathogenesis and presentations, diagnostic challenges, and risk factors for the disease. We hope that our revised manuscript based on all of the suggested edits make this objective much clearer.

Thank you all for you valuable time.

We congratulate the authors of this manuscript for the job done

Although this is just a general review of the literature which offer little benefit with respect to advancement of knowledge when compared to a systematic review with structured and more systematic collection of data, it has however provided reminder and useful evidence that could aid physician and other healthcare workers managing patients with Potts disease

Abstract

This need to be improved to better capture the main theme and message of the review article

Response: Thank you for this suggestion. We agree it needs to better convey the theme. We have made alterations to the abstract to better encapsulate the papers message as well as findings.

Introduction

One possible improvement for this introduction would be to provide more context and background information about Pott's disease, including its prevalence and impact on global health. It could also be helpful to provide a brief overview of the objectives and main points of the review, and to clearly define any important terms or concepts that will be discussed. Additionally, it might be useful to provide more specific information about the diagnostic challenges and risk factors associated with Pott's disease, and to explain how these factors contribute to delays in diagnosis and treatment. Finally, it could be helpful to provide some context for the relevance of this review, such as the need to improve awareness and understanding of Pott's disease in order to reduce morbidity and mortality.

Response: Thank you for the feedback. We agree and believe these items improve the quality of our paper. We have expanded upon the introduction section to increase the background and better set the stage of understanding for why our aims are significant.

Method

One possible improvement for the method section of this review article could be to provide more detail and clarity about the search strategy and inclusion/exclusion criteria. This could include specifying the specific databases that were searched (such as PubMed and NCBI), the time frame of the search (e.g. articles published in the past 10 years), and the specific keywords and MeSH terms that were used. It could also be helpful to provide more information about how the inclusion and exclusion criteria were applied, such as how studies were selected for inclusion and how duplicates were identified and removed. Additionally, it might be useful to provide some information about the characteristics of the studies that were included in the review, such as the study design, sample size, and patient population. This would help to provide more context and transparency about the studies that were included in the review and how they were selected. Also should also state here the period of the articles covered, quality assessment and how bias was addressed in article selection.

Response: Thank you, our method section certainly needed this. We have added timeline for our search as well as articles cited in paper and more inclusion and exclusion criteria.

Pathogenesis and Clinical Presentation of Pott’s Disease

One possible improvement for this section of the review article on the pathogenesis and clinical presentation of Pott's disease could be to provide more detail and clarity about the various presentations of spinal TB and the mechanisms by which they develop. This could include more information about the specific areas of the spine that are affected in each presentation and the specific characteristics and symptoms that are associated with each presentation. It could also be helpful to include more information about the diagnostic challenges and pitfalls that may be encountered when evaluating patients with Pott's disease, as well as any specific considerations that may need to be taken into account when formulating a treatment plan. Additionally, it might be useful to provide more information about the prognostic factors and outcomes associated with each presentation of spinal TB, including the likelihood of complications and disability and the overall effectiveness of different treatment approaches. Citing relevant studies and clinical guidelines could also be helpful in supporting the points made in this section of the review.

Response: Thank you. We have expanded upon this section to provide more data from relevant studies, as well as characteristics like risk. We have also included statistics on the different presentations.

Risk factors for Pott disease

Mentioning other risk factors for Pott's disease, such as older age, malnutrition, and previous TB infection

Response: We have included data regarding increased risk with age and malnutrition and also mentioned latent TB risk in the pathogenesis section.

Providing more information on the role of vitamin D deficiency in the development of Pott's disease, including mechanisms by which it may increase susceptibility

Response: We have expanded on the role of Vitamin D deficiency and speculated on mechanism of susceptibility.

Discussing the prevalence of Vitamin D deficiency in populations at risk for Pott's disease, such as those living in Sub-Saharan Africa

Response: We have added citation on the prevalence of vitamin D deficiency in Sub-Saharan Africa to tie in with our discussion on HIV as well.

Clarifying the relationship between estrogen deficiency and Pott's disease, including providing more information on the possible protective effect of estrogen on TB incidence

Response: Although little research is available regarding this connection, we have added speculation on the possible protective effect of estrogen on TB incidence.

Providing more detail on the link between HIV and Pott's disease, including how HIV infection may increase the risk of developing Pott's disease and how Pott's disease may contribute to HIV progression

Response: Thank you very much. We have elaborated further on the role of HIV and Pott’s disease, and vice versa.

Expanding on the role of genetics in the development of Pott's disease, including the potential role of other gene polymorphisms in addition to the FokI polymorphism in the VDR gene.

Response: Thank you for this suggestion. We were able to find an additional polymorphism in the VDR gene from 2021 that we speculate could potentially increase risk.

Response: Thank you for this suggestion. We were unable to find other links

Mentioning the impact of poverty and poor access to care on the risk of developing Pott's disease and the challenges these factors pose in diagnosis and treatment

Response: Thank you, we have included this in both this section, discussion, and introduction to tie this important point in to the theme of the paper.
